# Radiofrequency Cingulotomy as a Treatment for Incoercible Pain: Follow-Up for 6 Months

**DOI:** 10.3390/healthcare11192607

**Published:** 2023-09-22

**Authors:** Carlos Castillo Rangel, Gerardo Marin, Dylan Lucia Diaz Chiguer, Francisco Alberto Villegas López, Rodrigo Ramírez-Rodríguez, Alejandro Gómez Ibarra, Rosalba Aguilar-Velazquez, Julian Eduardo Soto Abraham

**Affiliations:** 1Department of Neurosurgery, “Hospital Regional 1° de Octubre”, Institute of Social Security and Services for State Workers (ISSSTE), Mexico City 07300, Mexico; neuro_cast27@yahoo.com; 2Neural Dynamics and Modulation Lab, Cleveland Clinic, Cleveland, OH 44195, USA; dra.dylandiaz@outlook.com (D.L.D.C.); xmarin14@gmail.com (A.G.I.); 3Department of Neurosurgery, “Hospital General de México”, Mexico City 06720, Mexico; dr.franciscovillegas@gmail.com (F.A.V.L.); gs17024572@egresados.uv.mx (J.E.S.A.); 4Brain Research Institute, Xalapa 91192, Mexico; zs18024964@estudiantes.uv.mx (R.R.-R.); rosaguilar@uv.mx (R.A.-V.)

**Keywords:** cingulotomy, neurosurgery, incoercible pain, analgesia, anterior cingulate cortex

## Abstract

Incoercible or intractable pain is defined as pain that is refractory to pharmacological treatment to such an extent that opioid and analgesic adverse effects outweigh the therapeutic effects. The anterior cingulate cortex (ACC) is involved in the perception of pain, especially emotional pain, so it is logical that cingulotomy has an effective therapeutic effect. Therefore, we evaluated the effectiveness of cingulotomy for the treatment of incoercible pain. An observational, longitudinal, retrospective, and analytical study was carried out on a series of cases in which bilateral cingulotomy was performed for incoercible pain, and follow-up was performed 6 months after neurosurgery in the outpatient clinic at the Neurotraumatology Clinic. A positive correlation was observed between pain intensity and medication use, indicating that an increase in pain was associated with a greater requirement for analgesics. The result was a significant reduction in pain, as measured by the visual analog scale of pain, and decreased drug use after cingulotomy. We concluded that cingulotomy reduces incoercible pain and the need for medication.

## 1. Introduction

Pain is defined as an unpleasant sensory and emotional experience, associated with actual or potential tissue injury according to the International Association for the Study of Pain [1]. Pain can be classified according to its duration: acute <3 months and chronic >3 months; pathogenesis: neuropathic (stimulus in central nervous system or pathways), nociceptive (according to the stimulated nociceptor, e.g., somatic or visceral), and psychogenic (it is characterized by having the need to increase the pharmacological dose); location: somatic (localized pain that is irradiated by nerve tracts) and visceral (poorly localized, irradiated to areas away from the pain); course: continuous (it does not disappear throughout the day) and irruptive (transient, even if they are pharmacologically well-controlled); intensity: mild (able to perform usual activities (VAS 1–3)), moderate (interferes with activities (VAS 4–6) (mild opioids must be used)), and severe (interferes with rest (VAS 7–10) (use of major opioids)); prognostic factor: good prognosis (visceral, bone or soft tissue pain, non-irruptive, without emotional distress, slow opioid scale, and without a history of alcoholism or drug addiction) and poor prognosis (neuropathic, irruptive pain, emotional distress, rapid opioid scale, and with a history of alcoholism and drug addiction); pharmacology: responsive to opioids (visceral and opioid pains), partially responsive to opioids (bone pain (NSAIDs are added) and compression of peripheral nerves (steroids are added)), and not responsive to opioids (pain due to spasm in striated muscles or damage in peripheral nerve (responds to antidepressants or anticonvulsants)) [2].

In a certain way, the usefulness of the above classification consists of pharmacological handling during a specific pain; for example, under severe intensity, major opioids are recommended, or under peripheral nerve damage, an anticonvulsant/antidepressant is preferred. However, it does not say what to do when it does not respond to pharmacological treatment, or which combinations are the most effective; for this, the WHO scale on pain handling is necessary (Figure 1), where invasive methods are mentioned in step 4, and among them, cingulotomy [2,3] and cingulotomy are considered for the treatment of incoercible pain in step IV of the WHO. This is defined as a section of fibers that reaches and exits the anterior cingulate gyrus [4]. Incoercible or intractable pain is defined as pain that is refractory to pharmacological treatment to such an extent that opioid and analgesic adverse effects outweigh therapeutic effects. Pain can be measured in three dimensions: sensory (pain intensity), emotional (unpleasant pain), and cognitive [5]. The anterior cingulate cortex (ACC) is involved in pain perception, especially in the emotional one, so it is logical for cingulotomy to have an effective therapeutic effect [6]. ACC is found in the medial part of the cerebral hemispheres around the corpus callosum. This structure is part of the limbic system and mainly has amygdala afferents and efferents towards the periaqueductal gray substance and the brainstem [5,6]. Ablation of this structure has not only been used for the treatment of incoercible pain but also for drug-refractory psychiatric diseases, including obsessive–compulsive disorder, depression, and severe anxiety [6,7]. It is believed that anterior cingulotomy does not modulate the sensation of pain intensity but the patient’s attention or emotional reaction to pain [8].

The effectiveness of cingulotomy is attributed to the cutting of nociceptive fibers from the thalamic nuclei [9]. This efficacy is supported by neuroimaging studies, which demonstrate a direct role of the cingulate gyrus in processing chronic pain, and by in vitro studies demonstrating long-term potentiation of excitatory synapses within the ACC surgery [10].

Although this technique has been studied for its therapeutic action against pain, it has fallen into disuse since the introduction of deep brain stimulation (DBS), spinal cord neuromodulation, and intrathecal therapy.

### Indications for Cingulotomy

Modern practices generally reserve cingulotomy as a second- or third-line treatment after non-injury surgery failures; however, cingulotomy should be the treatment of choice in some scenarios: cachexic or malnourished patients who cannot tolerate the implantation of a foreign body, patients with terminal cancer with short life expectancy, and patients who do not want to have or cannot afford deep brain stimulation.

It should be noted that ablative techniques offer the advantage of immediate pain relief; they do not require a routine medical examination or device maintenance in comparison to neuromodulative alternatives [9]. For these reasons, this study aims to assess the efficacy of cingulotomy by evaluating pain reduction and the decrease in analgesic consumption.

## 2. Materials and Methods

An observational, longitudinal, retrospective, and analytical study was carried out on a case series in which bilateral cingulotomy was performed due to incoercible pain. Follow-up was performed 6 months after hospitalization through outpatient clinic consultations at the Neurotraumatology Clinic between 2020 and 2022. All individuals were cared for by a multidisciplinary team comprising neurologists, neurosurgeons, psychiatrists, anesthesiologists, and neuropsychologists. Additionally, all patients were treated with Buprenorphine for pain management (Table 1). We observed headaches in three patients (patients 2, 5, and 6) as possible side effects. Nevertheless, the headaches disappeared at 6 months. All patients signed an informed consent.

### 2.1. Case Series

#### 2.1.1. Clinical Case 1

A 24-year-old male patient with a history of drug addiction every 2 days (cocaine, alcohol, and tobacco), accompanied with aggressive behavior, was repeatedly sent to rehabilitation centers, had inappropriate laughter attacks, generalized anxiety diagnosed by a psychiatrist after treatment with quetiapine, long-term epilepsy, which was controlled with clonazepam and magnesium valproate, and appendectomy without complications 3 years ago. He had a VAS score of 8/10 for headache, which is why he went in to receive service after being treated with Tramadol 1 tablet every 12 h, paracetamol 500 mg every 8 h, Ketorolac 30 mg every 12 h, and carbamazepine 1 tablet every 12 h. The pain of the headache was classified as chronic, psychogenic, somatic, and irruptive, ameliorated due to drug use, thus categorizing the pain with a poor prognosis, after which the decision was made to intervene by performing bilateral anterior cingulotomy (Figure 2), which was carried out without complications and immediately decreased the pain to a VAS score of 1/10. After the one-month follow-up, he had the same VAS score of 1/10, and the same medications were continued except for carbamazepine.

#### 2.1.2. Clinical Case 2

The second case involved a 72-year-old female patient with no history of DM2 and HTN, with obesity, controlled hyperlipidemia, hypothyroidism treated with levothyroxine, was diagnosed with major depression and obsessive–compulsive disorder treated with psychiatry, and currently has asymptomatic structural epilepsy. She underwent surgical procedures at the C3, C4, and C5 levels (she has no reasoning recollection) and resection of the cavernoma (which left her with incoercible vertigo), has arteriovenous malformation (AVM) that left her with incoercible vertigo sequela, and has pacemakers placed via ventricular fibrillation, the surgical procedures of which had no apparent complications. She was admitted with cervial pain with a VAS score of 9/10, and her mood swings were being treated with vortioxetine, levetiracetam, trevinn, pregabalin, ketoprofen, ketorolac, and tramadol; therefore, it was decided to perform a cingulotomy to manage the pain. The uncomplicated cingulotomy (Figure 3) resulted in a VAS score of 1/10, and the pain was subsequently managed by taking pregabalin 75 mg at night and ketorolac in case of pain.

#### 2.1.3. Clinical Case 3

A 71-year-old female patient had a history of Arnold–Chiari malformation, which was operated on 22 years ago without complications, hernioplasty 10 years ago, cesarean section 35 years ago, with sleep apnea for the last 2 years treated with oxygen at night with CPAP, and occasional marijuana use 1 year ago. She was admitted due to pain starting with VAS 2/10, which quickly increased to a VAS score of 10/10, treated with tramadol + paracetamol (Zaldiar) 1 tablet every 8 h, sublingual ketorolac 10 mg 1 tablet every 8 h, pregabalin 1 tablet every 24 h, gabapentin 300 mg 1 tablet at night, and magnesium valproate 200 mg 1 tablet every 12 h. However, due to no symptom improvement, it was decided to intervene by performing a cingulotomy (Figure 4), and the neurosurgery was carried out without complications. At 3 months follow-up, she mentioned that she was asymptomatic. Six months later, she continued without pain.

#### 2.1.4. Clinical Case 4

A 62-year-old female patient had a history of Systemic Arterial Hypertension (SAH), which was controlled with 1 tablet of losartan and spironolactone every 24 h. She was admitted due to failed back syndrome and cervicalgia from disc hernias (7 vertebroplasty surgeries in 2002) that caused generalized quadriparesis and spondyloarthrosis. She had fibromyalgia with chronic pain, which was sometimes psychogenic and at times nociceptive and persisted despite treatment with acetaminophen 500 mg 1 tablet every 8 h, tramadol 1 tablet every 12 h, celecoxib 200 mg 1 tablet every 8 h, gabapentin 300 mg 1 tablet every 12 h, sublingual buprenorphine (which was not tolerated due to allergy), and 1 fentanyl patch every 72 h. As there was no improvement in the clinical profile, it was decided to opt for cingulotomy in order to treat the symptoms. The neurosurgery was carried out without complications (Figure 5), which resulted in an improvement with a VAS score of 3/10 one month after surgery; she was later prescribed lysine clonixinate 1 tablet 400 mg every 24 h, lamictal (lamotrigine) 25 mg 1 tablet every 24 h, and zolpidem 10 mg 1 tablet at night to fall asleep.

#### 2.1.5. Clinical Case 5

A 60-year-old female patient had a history of generalized chronic pain with a VAS score of 10/10 proclivity for the head, a history of tension headache, jaw stiffness, which was treated with rhizolysis, and drug blockade prior to bilateral cingulotomy, which was treated with xummer (etoricoxib) 90 mg 1 capsule every 12 h, ketorolac 10 mg 1 tablet every 12 h, duloxetine 60 mg 1 tablet every 12 h, gabapentine 300 mg 1 tablet every 12 h, zaldiar (tramadol and paracetamol) 1 tablet every 8 h, lamotigrine 100 mg 1 tablet every 12 h, and prednisone 5 mg 1 tablet every 24 h. After the bilateral cingulotomy (Figure 6), only xummer (Etoricoxib), duloxetine, gabapentine, ketorolac, and zaldiar (tramadol and paracetamol) were continued.

#### 2.1.6. Clinical Case 6

A 63-year-old female patient had a history of a VAS score of 10/10 due to mixed polyneuropathy in the legs, smoking, cannabis use, SLE, osteoarthritis, COPD, catheterization due to AMI, and seizures during her intrathecal morphine pump infusion treatment for 6 years. Her surgical history was as follows: 2 “aesthetic” facial reconstruction surgeries at age 32, cross ligament reconstruction at age 28, BTO at age 35, cholecystectomy 15 years ago, cardiac catheterization 6 years ago due to AMI, morphine pump infusion placement in 2003, morphine pump removal in 2019 under cannabis treatment with 2 cigarettes daily (self-medicated), prednisone 5 mg every 12 h, pregabalin 150 mg every 12 h, acemetacin 60 mg 1 tablet every 24 h, losartan 100 mg every 12 h, metoprolol 100 mg every 12 h, carbamazepine 200 mg 2 tablets every 8 h, clonazepam 1 tablet every 12 h, celecoxib 100 mg 1 tablet every 8 h, ketoprofen 100 mg orally every 12 h, ibuprofen 800 mg orally every 8 h, tramadol 100 mg orally every 8 h, amlodipine 5 mg orally every 24 h, senosidos ab 2 tablets every 24 h, omeprazole 20 mg orally every 12 h, magnesium valproate 400 mg orally every 12 h, and indomethacin 25 mg orally every 24 h. After surgery, only morphine infusion (Figure 7) with sublingual ketorolac was continued.

### 2.2. Surgical Procedure

Under the effects of anesthesia, the head frame was first placed and then helical computed tomography and magnetic resonance imaging were taken in order to plan the coordinates, after which trephination was performed, dura mater was coagulated, the cingulum was reached with “inomed” electrodes and radiofrequency lesion (we used “Cosman Medical” radiofrequency generator), which was performed at 90 s and 90° Celsius, obtaining an impedance of 390–620 ohms; this procedure was performed bilaterally. Three lesion sites were examined bilaterally on two levels (depending on the patient), which were performed in the anterior and posterior targets, as described below.

### 2.3. Statistical Analysis

Poisson regression tests were used to assess the differences in medication and VAS scores before and after surgical intervention. The category “Pre-intervention surgery” was used as the reference level for each respective model. The results were reported as incidence rate ratios (IRR) with their respective 95% confidence intervals. Overdispersion was not observed in either model. In addition, Spearman’s correlation coefficient was calculated to measure the strength of the association between medication and VAS scores as follows: rho = 0.1–0.2 (poor), 0.3–0.5 (fair), 0.6–0.7 (moderate), and 0.8–0.9 (very strong) [11,12]. For all analyses, the significance level was set at *p* < 0.05. Statistical analysis was performed using R studio v.4.1.3, and data visualization was carried out using JASP 0.17, both for Mac.

## 3. Results

### 3.1. Demographic and Clinical Data

Six patients took part in this study, and none of them experienced relapses after the cingulotomy surgery (Table 2).

### 3.2. Patients Exhibit Lower Medication Scores after Cingulotomy

After the intervention, patients scored lower on medication score (IRR = 0.39), indicating a 61% reduction in the probability of medication consumption due to the intervention (Table 3) (Figure 8).

### 3.3. Patients Exhibit Lower VAS Scores after Cingulotomy

After the intervention, patients scored lower on the VAS score (IRR = 0.18), indicating an 82% reduction in pain perception due to the intervention (Table 3) (Figure 9).

### 3.4. Visual Pain Perception Positively Associates with Medication Scores

Spearman’s correlation between medication and VAS score was significant (*p* < 0.05), reflecting a very strong association (rho = 0.83).

## 4. Discussion

The objective of this study was to determine the effectiveness of bilateral cingulotomy in decreasing chronic incoercible pain, which was demonstrated in the results by a decrease in the analogous visual scale score and by a decrease in the need for medication. This was demonstrated through a correlation whereby the greater the medication used, the greater the pain scale; therefore, a decrease in medication should be considered as a decrease in pain. None of the patients were found to have relapsed during the 6-month follow-up at the outpatient clinic.

### WHO Scale

Prior to the surgery, perhaps after experiencing incoercible pain, the patients went to see many physicians without being able to keep the pain under control; therefore, it seems that almost none of them received adequate pain relief as indicated by the WHO, given that sometimes they jumped from step 1 to step 3 (this observation was gathered through the anamneses obtained from the medical records). This is possibly attributed to the fact that the drugs had little or no effectiveness in pain management, and in this scenario, by following their best intentions, previous physicians wanted rapid improvement of the patients. It should be noted that by going to multiple doctors without adequate follow-ups, the patients ended up self-medicating a combination of multiple drugs; two of the patients even self-administered marijuana, and a third patient was immersed in drug addiction (Table 4).

The patients in this study had a 6-month follow-up after cingulotomy, in which they were evaluated to determine the effect of neurosurgery on pain, which coincided with two patients who were also involved in a study by Deng 2019 et al., where the effectiveness of cingulotomy continued to decrease the pain even after 12 months [13].

The main limitation of the study is the sample size for data analysis, as well as the fact that patient follow-up was not continued over time due to the cost and time; however, this does not diminish the effectiveness of cingulotomy in managing incoercible pain.

## 5. Conclusions

Bilateral cingulotomy decreases incoercible pain, which was measured after 6 months according to the visual analog scale, and likewise, it decreases the need for pain medication along with its adverse effects.

## Figures and Tables

**Figure 1 healthcare-11-02607-f001:**
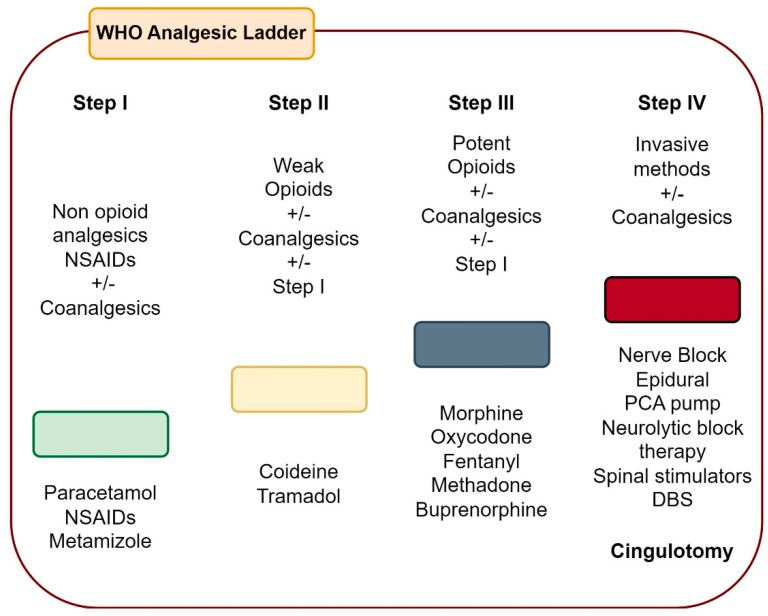
Analgesic Ladder World Health Organization (WHO) [2,3,4].

**Figure 2 healthcare-11-02607-f002:**
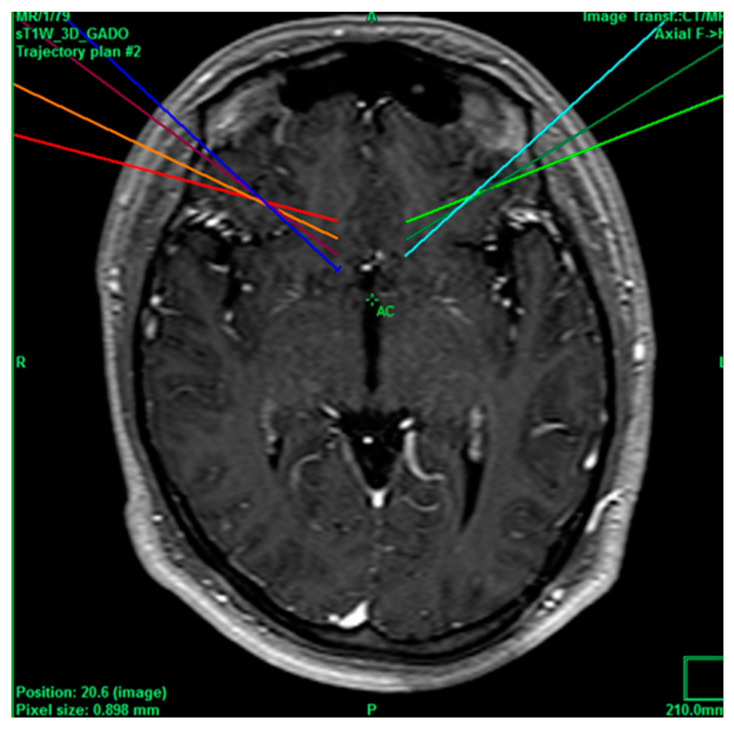
Bilateral projection of cross-sectional cingulotomy from patient 1.

**Figure 3 healthcare-11-02607-f003:**
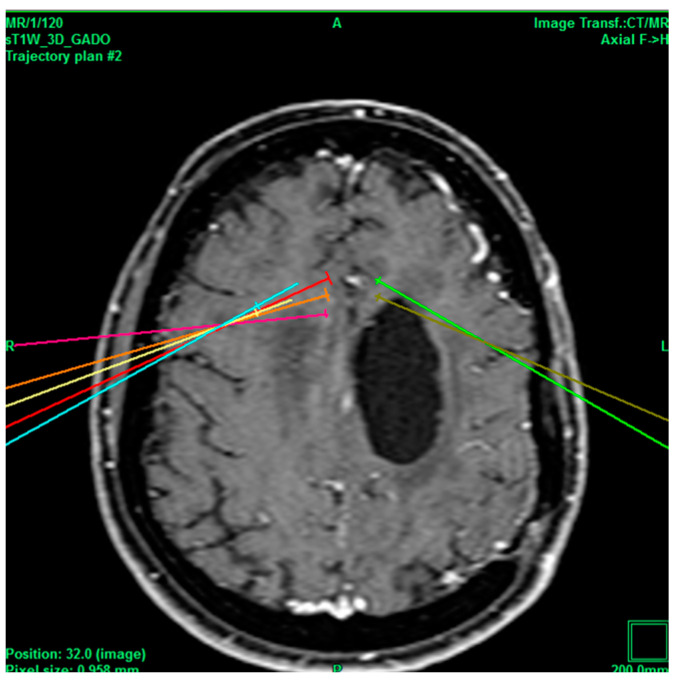
Bilateral projection of cross-sectional cingulotomy from patient 2. Note hypodense region due to cavernoma.

**Figure 4 healthcare-11-02607-f004:**
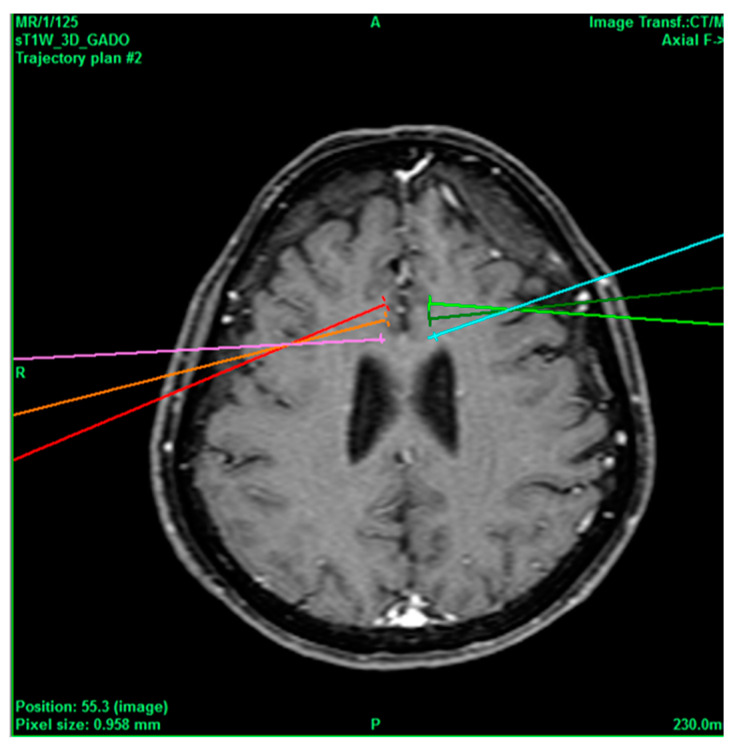
Bilateral projection of cross-sectional cingulotomy from patient 3.

**Figure 5 healthcare-11-02607-f005:**
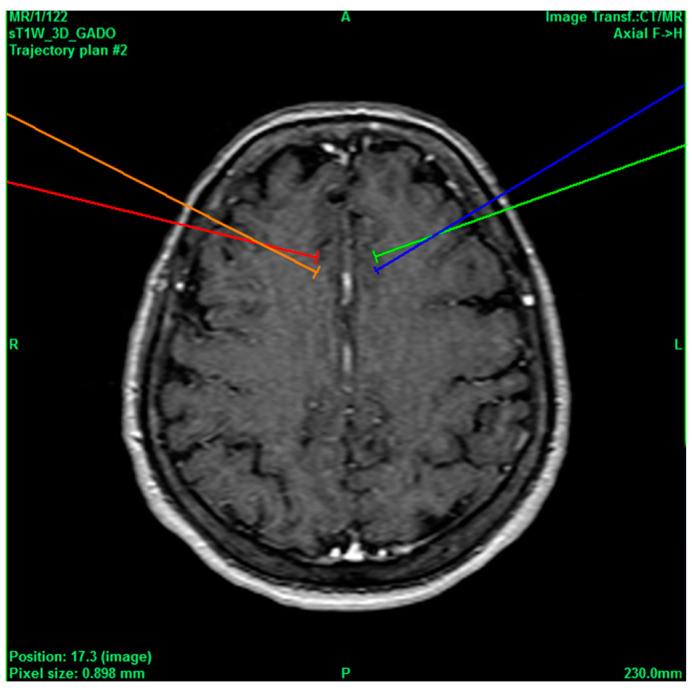
Bilateral projection of cross-sectional cingulotomy from patient 4.

**Figure 6 healthcare-11-02607-f006:**
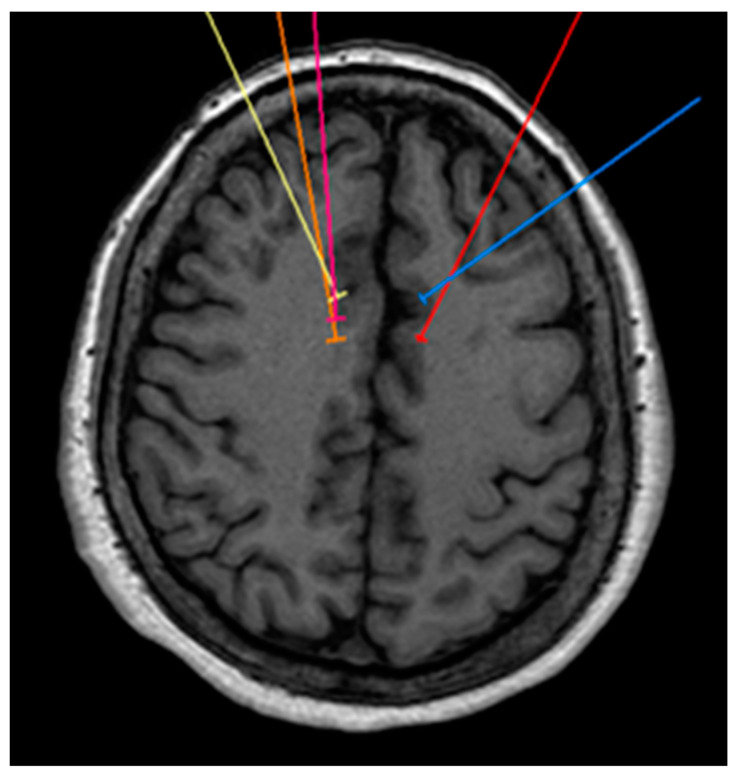
Bilateral projection of cross-sectional cingulotomy from patient 5.

**Figure 7 healthcare-11-02607-f007:**
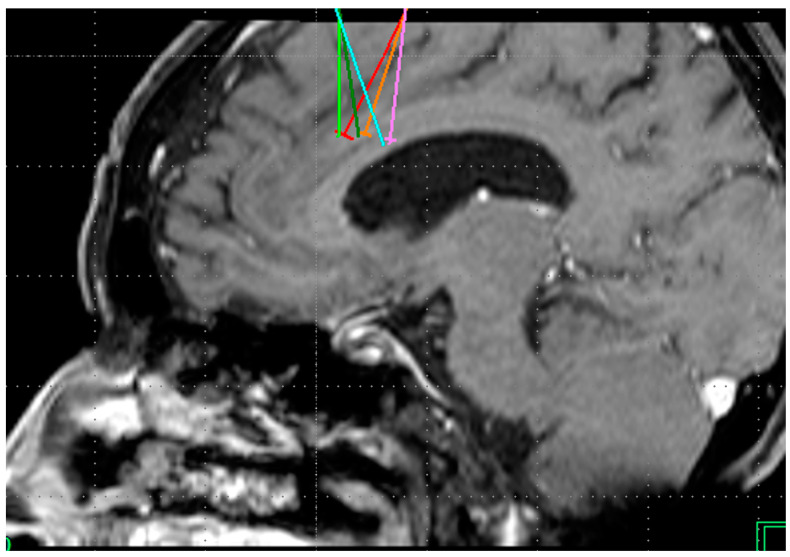
Projection of cingulotomy from patient 6.

**Figure 8 healthcare-11-02607-f008:**
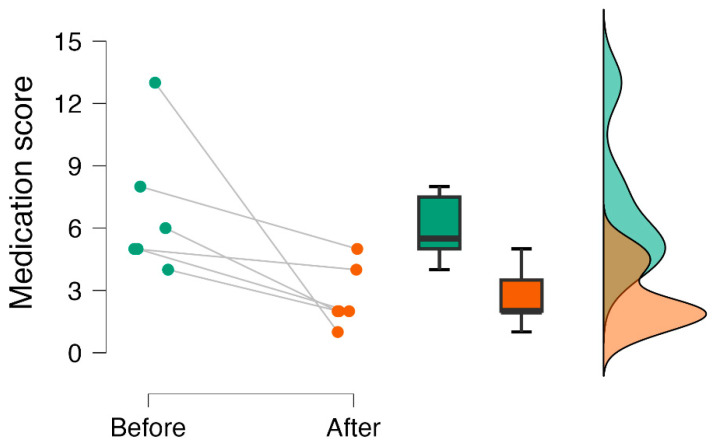
Raincloud plot of medication score before and after cingulotomy.

**Figure 9 healthcare-11-02607-f009:**
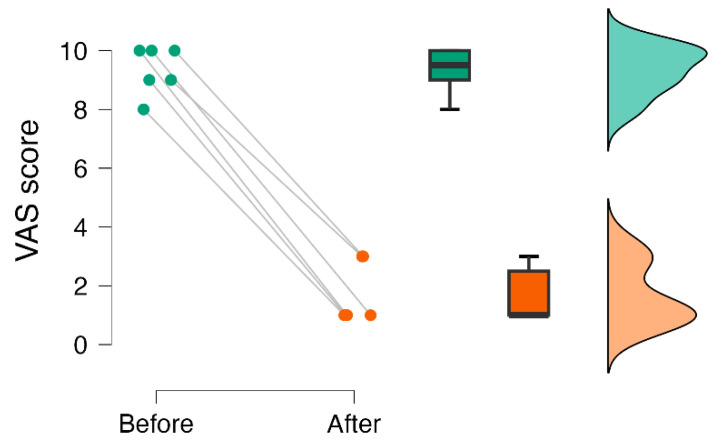
Raincloud plot of VAS score before and after cingulotomy.

**Table 1 healthcare-11-02607-t001:** Analgesic ladder applied to patients with cingulotomy.

Patient	Step 1	Step 2	Step 3
1	NSAIDs + Coanalgesics	Tramadol Buccal 100 mg c/12	Buprenorphine Buccal 900 mcg c/12 h
2	NSAIDs + Coanalgesics	Tramadol Buccal 100 mg c/12	Buprenorphine Buccal 900 mcg c/12 h
3	NSAIDs + Coanalgesics	Tramadol + Paracetamol Buccal 37.5/325 mg c/12 h	Buprenorphine Buccal 900 mcg c/12 h
4	NSAIDs + Coanalgesics	Tramadol Buccal 100 mg c/12	Buprenorphine (sublingual)/Fentanyl patches
5	NSAIDs + Coanalgesics	Tramadol + Paracetamol Buccal 37.5/325 mg c/12 h	Buprenorphine Buccal 900 mcg c/12 h
6	NSAIDs + Coanalgesics	Tramadol Buccal 100 mg c/12	Buprenorphine Buccal 900 mcg c/12 h

**Table 2 healthcare-11-02607-t002:** Demographic and clinical characteristics of the patients.

Gender, n (%)	F, 5 (83.33): M, 1 (16.66)
Age, M (SD)	F (65.6 ± 5.5): M (NA)
**Pathogeny, n (%)**	
Psychogenic	3 (50)
Nociceptive	1 (16.66)
Psychogenic/Nociceptive	2 (33.33)
**Location of pain, n (%)**	
Somatic	5 (83.33)
Visceral	1 (16.66)
**Course of symptoms, n (%)**	
Irruptive	2 (33.33)
Continuous	4 (66.66)
**Post-surgery relapse after six months, n (%)**	
Yes	0 (0%)
No	6 (100%)

**Table 3 healthcare-11-02607-t003:** IRR = incidence rate ratio; SE: standard error; 95% CI; 95% confidence interval. VAS: visual analogue scale.

Intervention	IRR (95% CI)	SE	*p* Value
Post-intervention medication score	0.39 (0.21–0.68)	0.12	0.001
Post-intervention VAS score	0.18 (0.09–0.33)	0.06	*p* < 0.001

**Table 4 healthcare-11-02607-t004:** Description of the cases.

Cases	Gender	Age	Etiology of Pain	Background	Duration	Pathogenesis	Location	Course	Prognosis	# Pre Med	# Post Med	PRE VAS	POST VAS	Relapse (6 Months)
1	M	24	Cephalea	Drug addiction + Epilepsy + Refractory aggressiveness	Chronic	Doxogenic	Somatic	Irruptive	Poor	5	4	8	1	No
2	F	72	Cervicalgia	AVM PO + OCD + Cavernoma PO + Major depression + Epilepsy + CVD PO + Incoercible vertigo	Chronic	Doxogenic	Somatic	Continuous	Poor	4	2	9	1	No
3	F	71	Generalized chronic pain	Arnold–Chiari + Sleep apnea + Major depression + OCD	Chronic	Doxogenic	Visceral	Irruptive	Poor	5	2	10	1	No
4	F	62	Failed Back Syndrome/Generalized Spondyloarthrosis	SAH + Fibromyalgia + Cervical hernia PO	Chronic	Nociceptive/Psychogenic	Somatic	Continuous	Poor	6	2	9	3	No
5	F	60	Generalized chronic pain/Headache	Tension headache + Rizolysis + Jaw stiffness + Drug blockade	Chronic	Nociceptive	Somatic	Continuous	Poor	8	5	10	3	No
6	F	63	Mixed polyneuropathy in legs	Smoking + Cannabis use + SLE + Osteoarthritis + COPD + Mixed Polyneuropathy + Catheterization due to AMI + Seizures + Intrathecal morphine pump infusion for 6 years	Chronic	Nociceptive/Psychogenic	Somatic	Continuous	Poor	13	1	10	1	No

## Data Availability

The data presented in this study are available in Table 3.

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
