# Peer review of "Radiofrequency Cingulotomy as a Treatment for Incoercible Pain: Follow-Up for 6 Months"

_healthcare, 2023, doi:10.3390/healthcare11192607_

Round 1

Reviewer 1 Report

The study presents a small series of patients who underwent bilateral cingulotomy for pain relief. The work is rich in data but there are some concerns that need to be resolved to make it acceptable for publication 1) In the title it should be specified that the follow-up for the evaluation of efficacy is only 6 months. 2) In the description of the cases, reference is made to the recommendations of the WHO analgesic latter which provides for neurosurgical interventions only after having reached grade III and not having responded adequately to this therapy. In the reports, cases 1, 2, 3, 4 and 5 had not reached grade 3 which involves the use of major opioids. So neurosurgical intervention had no formal indication. This point needs to be clarified and discussed. 3) In table 2 : the description of the number of analgesics use before and after shows a significant difference only in one case, while others cases do not seem to have a reduction in the consumption of analgesics. This point needs to be discussed. On the other hand, a reduction of the VAS seems to have been demonstrated after the intervention, at least for 6 months. 4) No undesirable effects after the intervention are reported. This is quite wonderful and this data should be verified. Minor concerns are: It is not clear in which hospital (NT) the patients have been treated Line 39= “enolism” what means? Maybe alcoholism?

It is sufficient

Author Response

  1. We added in the title the patient’s follow-up.
  2. Since we gave buprenorphine to all, patients are in step 3. It wasn’t sufficient for pain relief, so we continued to step 4. You can see this sentence in row 119.
  3. We can’t interpret a significant difference with only the values. We used the incidence rate ratio for this measure.
  4. We added side effects in row 120 (headache). We added the name of the Neurotraumatology Clinic. We changed the word “alcoholism” instead of “enolism”.

Thank you so much for your time.

Reviewer 2 Report

This study reported a case series of patients who underwent bilateral cingulotomy because of uncontrollable pain.

Title: The "case series" should be mentioned in the title.

Introduction:
Figure 1: Did the authors use Figure 1 with permission from WHO?

Methods:

Did patients sign consent?

Did case 1 receive step 3 of the analgesic ladder?

"The same medications were kept except for carbamazepine." If the patient was pain-free, why did he receive similar medications?

Did Case 2 receive the first 3 steps of the analgesic ladder? Similarly, for all cases, please report the steps of the analgesic ladder that patients received.

"radiofrequency lesion was performed." Please mention the method of surgery in the title.

What company manufactured the "radiofrequency device," and did the authors receive any grants or support from that company?

References:

Please provide an English translation for the titles of references that are not in English in the [].

Author Response

  1. The phrase “Case report” is above the title.
  2. We don’t have special permission but refer to the citations where the scale is mentioned. We created the figure with that information.
  3. When patients enter the clinic, they sign an informed consent.
  4. Since patient 3 received buprenorphine, she and all patients are in step 3 (row 119). We added a label (row 120).
  5. We discussed that the follow-up was for 6 months maybe could decrease the medication more at 12 months (Deng 2019). It should be noted that inference result analysis must be done with statistics. Our study used an incidence rate ratio (0.39) to see this phenomenon.
  6. We added the word “radiofrequency” in the title.
  7. We translated the Spanish references into English.

Round 2

Reviewer 1 Report

I the present form the article is suitable for publication

Author Response

I appreciate your help.

Reviewer 2 Report

I appreciate your responses to my inquiries. Kindly address my final suggestions:

For all cases, please report the steps of the analgesic ladder that patients received in a table.‎

Which company produced the "radiofrequency device," and did the authors receive any grants or assistance from said company? Please add the answer to this question to the manuscript.

Author Response

We added Table 2 with the steps of the analgesic ladder scale. We don't have any conflict of interest, and we don't have any grants or assistance.
